# Lossless Compression of Structured Convolutional Models via Lifting

**Gustav Sourek, Filip Zelezny, Ondrej Kuzelka**
Department of Computer Science
Czech Technical University in Prague
`{souregus,zelezny,kuzelon2}@fel.cvut.cz`

## Abstract

Lifting is an efficient technique to scale up graphical models generalized to relational domains by exploiting the underlying symmetries. Concurrently, neural models are continuously expanding from grid-like tensor data into structured representations, such as various attributed graphs and relational databases. To address the irregular structure of the data, the models typically extrapolate on the idea of convolution, effectively introducing parameter sharing in their, dynamically unfolded, computation graphs. The computation graphs themselves then reflect the symmetries of the underlying data, similarly to the lifted graphical models. Inspired by lifting, we introduce a simple and efficient technique to detect the symmetries and compress the neural models without loss of any information. We demonstrate through experiments that such compression can lead to significant speedups of structured convolutional models, such as various Graph Neural Networks, across various tasks, such as molecule classification and knowledge-base completion.

## 1 Introduction

Lifted, often referred to as *templated*, models use highly expressive representation languages, typically based in weighted predicate logic, to capture symmetries in relational learning problems (Koller et al., 2007). This includes learning from data such as chemical, biological, social, or traffic networks, and various knowledge graphs, relational databases and ontologies. The idea has been studied extensively in probabilistic settings under the notion of lifted graphical models (Kimmig et al., 2015), with instances such as Markov Logic Networks (MLNs) (Richardson & Domingos, 2006) or Bayesian Logic Programs (BLPs) (Kersting & De Raedt, 2001).

In a wider view, *convolutions* can be seen as instances of the templating idea in neural models, where the same parameterized pattern is being carried around to exploit the underlying symmetries, i.e. some forms of shared correlations in the data. In this analogy, the popular Convolutional Neural Networks (CNN) (Krizhevsky et al., 2012) themselves can be seen as a simple form of a templated model, where the template corresponds to the convolutional filters, unfolded over regular spatial grids of pixels. But the symmetries are further even more noticeable in structured, relational domains with discrete element types. With convolutional templates for regular trees, the analogy covers Recursive Neural Networks (Socher et al., 2013), popular in natural language processing. Extending to arbitrary graphs, the same notion covers works such as Graph Convolutional Networks (Kipf & Welling, 2016) and their variants (Wu et al., 2019), as well as various Knowledge-Base Embedding methods (Wang et al., 2017). Extending even further to relational structures, there are works integrating parameterized relational logic templates with neural networks (Sourek et al., 2018; Rocktäschel & Riedel, 2017; Marra & Kuželka, 2019; Manhaeve et al., 2018).

The common underlying principle of templated models is a joint parameterization of the symmetries, allowing for better generalization. However, standard lifted models, such as MLNs, provide another key advantage that, under certain conditions, the model computations can be efficiently carried out without complete template unfolding, often leading to even exponential speedups (Kimmig et al., 2015). This is known as "lifted inference" (Kersting, 2012) and is utilized heavily in lifted graphical models as well as database query engines (Suciu et al., 2011). However, to our best knowledge, this idea has been so far unexploited in the neural (convolutional) models. The main contribution of

this paper is thus a "lifting" technique to compress symmetries in convolutional models applied to structured data, which we refer to generically as "structured convolutional models".

## 1.1 RELATED WORK

The idea for the compression is inspired by lifted inference (Kersting, 2012) used in templated graphical models. The core principle is that all equivalent sub-computations can be effectively carried out in a single instance and broadcasted into successive operations together with their respective multiplicities, potentially leading to significant speedups. While the corresponding "liftable" template formulae (or database queries) generating the isomorphisms are typically assumed to be given (Kimmig et al., 2015), we explore the symmetries from the unfolded ground structures, similarly to the approximate methods based on graph bisimulation (Sen et al., 2012). All the lifting techniques are then based in some form of first-order variable elimination (summation), and are inherently designed to explore *structural* symmetries in graphical models. In contrast, we aim to additionally explore *functional* symmetries, motivated by the fact that even structurally different neural computation graphs may effectively perform identical function.

The learning in neural networks is also principally different from the model counting-based computations in lifted graphical models in that it requires many consecutive evaluations of the models as part of the encompassing iterative training routine. Consequently, even though we assume to unfold a complete computation graph before it is compressed with the proposed technique, the resulting speedup due to the subsequent training is still substantial. From the deep learning perspective, there have been various model compression techniques proposed to speedup the training, such as pruning, decreasing precision, and low-rank factorization (Cheng et al., 2017). However, to our best knowledge, the existing techniques are lossy in nature, with a recent exception of compressing ReLU networks based on identifying neurons with linear behavior (Serra et al., 2020). None of these works exploit the model computation symmetries. The most relevant line of work here are Lifted Relational Neural Networks (LRNNs) (Sourek et al., 2018) which however, despite the name, provide only templating capabilities without lifted inference, i.e. with complete, uncompressed ground computation graphs.

## 2 BACKGROUND

The compression technique described in this paper is applicable to a number of structured convolutional models, ranging from simple recursive (Socher et al., 2013) to fully relational neural models (Sourek et al., 2018). The common characteristic of the targeted learners is the utilization of convolution (templating), where the same parameterized pattern is carried over different sub-parts of the data (representation) with the same local structure, effectively introducing repetitive sub-computations in the resulting computation graphs, which we exploit in this work.

### 2.1 GRAPH NEURAL NETWORKS

Graph neural networks (GNNs) are currently the most prominent representatives of structured convolutional models, which is why we choose them for brevity of demonstration of the proposed compression technique. GNNs can be seen as an extension of the common CNN principles to completely irregular graph structures. Given a particularly structured input sample graph $S_j$, they dynamically unfold a multi-layered computation graph $\mathcal{G}_j$, where the structure of each layer $i$ follows the structure of the whole input graph $S_j$. For computation of the next layer $i+1$ values, each node $v$ from the input graph $S_j$ calculates its own value $h(v)$ by aggregating $A$ ("pooling") the values of the adjacent nodes $u : edge(u,v)$, transformed by some parametric function $C_{W_1}$ ("convolution"), which is being reused with the same parameterization $W_1$ within each layer $i$ as:

$$\tilde{h}(v)^{(i)} = A^{(i)}(\{C_{W_1^i}^{(i)}(h(u)^{(i-1)})|u : edge(u,v)\}) \tag{1}$$

The $\tilde{h}^{(i)}(v)$ can be further combined through another $C_{W_2}$ with the central node's representation from the previous layer to obtain the final updated value $h^{(i)}(v)$ for layer $i$ as:

$$h(v)^{(i)} = C_{W_2^i}^{(i)}(h(v)^{(i-1)}, \tilde{h}(v)^{(i)}) \tag{2}$$

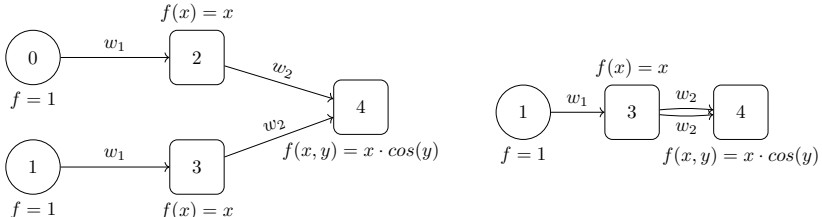

Figure 1: Depiction of the computation graph (left) compression (right) from Example 1.

This general principle covers a wide variety of GNN models, such the popular GCNs (Kipf & Welling, 2016), graph-SAGE (Hamilton et al., 2017), GIN (Xu et al., 2018a), and others (Xu et al., 2018b; Gilmer et al., 2017), which then reduces to the respective choices of particular aggregations $A$ and transformations $C_W$. An example computation graph of a generic GNN unfolded over an example molecule of methane is shown in Fig. 2.

## 2.2 COMPUTATION GRAPHS

For the sake of this paper, let us now define the common notion of a computation graph more formally. A computation graph is a tuple $\mathcal{G} = (\mathcal{N}, \mathcal{E}, \mathcal{F})$, where $\mathcal{N} = (1, 2, \ldots, n)$ is a list of *nodes* and $\mathcal{E} \subseteq \mathcal{N}^2 \times \mathbb{N}$ is a list of directed labeled *edges*. Each labeled edge is a triple of integers $(n_1, n_2, l)$ where $n_1$ and $n_2$ are nodes of the computation graph and $l$ is the *label*. The labels are used to assign weights to the edges in the computation graph. Note this allows to define the weight sharing scheme as part of the graph (cf Example 1 below). Finally, $\mathcal{F} = \{f_1, f_2, \ldots, f_n\}$ is the list of *activation functions*, one for each node from $\mathcal{N}$. As usual, the graph is assumed to be acyclic. Children of a node $N$ are naturally defined as all those nodes $M$ such that $(M, N, L) \in \mathcal{E}$, and analogously for parents. Note that since $\mathcal{E}$ is a list, edges contained in it are ordered, and the same edge may appear multiple times (which will be useful later). Children of each node are also ordered – given two children $C$ and $C'$ of a node $N$, $C$ precedes $C'$ iff $(C, N, L)$ precedes $(C', N, L')$ in the list of edges $\mathcal{E}$. We denote the lists of children and parents of a given node $N$ by *Children*$(N)$ and *Parents*$(N)$, respectively. Computation graphs are then evaluated bottom up from the leaves of the graph (nodes with no children) to the roots of the graph (nodes with no parents). Given a list of weights $\mathcal{W}$, we can now define the *value* of a node $N \in \mathcal{N}$ recursively as:

$$value(N; \mathcal{W}) = f_N\Big(\mathcal{W}_{L_1} \cdot value(M_1; \mathcal{W}), \ldots, \mathcal{W}_{L_m} \cdot value(M_m; \mathcal{W})\Big),$$

where $(M_1, \ldots, M_m) \equiv$ *Children*$(N)$ is the (ordered) list of children of the node $N$, and $L_1, \ldots, L_m$ are the labels of the respective edges $(M_1, N, L_1), \ldots, (M_m, N, L_m) \in \mathcal{E}$, and $\mathcal{W}_{L_i}$ is the $L_i$-th component of the list $\mathcal{W}$. Note that with the structured convolutional models, such as GNNs, we assume dynamic computation graphs where each learning sample $S_j$ generates a separate $\mathcal{G}_j$. Consequently, we can associate the leaf nodes in each $\mathcal{G}_j$ with constant functions[1], outputting the corresponding node (feature) values from the corresponding structured input sample $S_j$.

## 3 PROBLEM DEFINITION

The problem of detecting the symmetries in computation graphs can then be formalized as follows.

**Definition 1** (Problem Definition). *Let $\mathcal{G} = (\mathcal{N}, \mathcal{E}, \mathcal{F})$ be a computation graph. We say that two nodes $N_1, N_2$ are equivalent if, for any $\mathcal{W}$, it holds that value$(N_1; \mathcal{W}) = $ value$(N_2; \mathcal{W})$. The problem of detecting symmetries in computation graphs asks to partition the nodes of the computation graph into equivalence classes of mutually equivalent nodes.*

**Example 1.** *Consider the computation graph $\mathcal{G} = (\mathcal{N}, \mathcal{E}, \mathcal{F})$, depicted in Fig. 1, where*

$$\mathcal{N} = \{0, 1, 2, 3, 4\}, \ \mathcal{E} = ((0, 2, 1), (1, 3, 1), (2, 4, 2), (3, 4, 2)),$$
$$\mathcal{F} = \{f_0 = f_1 = 1, f_2(x) = f_3(x) = x, f_4(x, y) = x \cdot cos(y)\}.$$

---

[1]in contrast to static computation graphs where these functions are identities requiring the features at input.

*Let $\mathcal{W} = (w_1, w_2)$ be the weight list. The computation graph then computes the function $(w_1 w_2) \cdot cos(w_1 w_2)$. It is not difficult to verify that the nodes $\{0, 1\}$, and $\{2, 3\}$ are functionally equivalent. This also means, as we discuss in more detail in the next section, that we can "merge" them without changing the function that the graph computes. The resulting reduced graph then has the form*

$$\mathcal{N} = \{1, 3, 4\}, \ \mathcal{E} = \{(1, 3, 1), (3, 4, 2), (3, 4, 2)\},$$
$$\mathcal{F} = \{f_1 = 1, f_3(x) = x, f_4(x, y) = x \cdot cos(y)\}.$$

In the example above, the nodes $\{0, 1\}$ and $\{2, 3\}$ are in fact also isomorphic in the sense that there exists an automorphism (preserving weights and activation functions) of the computation graph that swaps the nodes. Note that our definition is less strict: all we want the nodes to satisfy is *functional* equivalence, meaning that they should evaluate to the same values for any initialization of $\mathcal{W}$.

We will also use the notion of *structural-equivalence* of nodes in computational graphs. Two nodes are structurally equivalent if they have the same outputs for any assignment of weights $\mathcal{W}$ and for any replacement of any of the activation functions in the graph.[2] That is if two nodes are structurally equivalent then they are also functionally equivalent but not vice versa. Importantly, the two nodes do not need to be automorphic[3] in the graph-theoretical sense while being structurally equivalent, which also makes detecting structural equivalence easier from the computational point of view. In particular, we describe a simple polynomial-time algorithm in Section 4.2.

## 4 Two Algorithms for Compressing computation Graphs

In this section we describe two algorithms for compression of computation graphs: a non-exact algorithm for compression based on functional equivalency (cf. Definition 1) and an exact algorithm for compression based on detection of structurally-equivalent nodes in the computation graph. While the exact algorithm will guarantee that the original and the compressed computation graphs represent the same function, that will not be the case for the non-exact algorithm. Below we first describe the non-exact algorithm and then use it as a basis for the exact algorithm.

### 4.1 A Non-Exact Compression Algorithm

The main idea behind the non-exact algorithm is almost embarrassingly simple. The algorithm first evaluates the computation graph with $n$ randomly sampled parameter lists $\mathcal{W}_1, \ldots, \mathcal{W}_n$, i.e. with $n$ random initializations of the (shared) weights, and records the values of all the nodes of the computation graph (i.e. $n$ values per node). It then traverses the computation graph from the output nodes in a breadth-first manner, and whenever it processes a node $N$, for which there exists a node $N'$ that has not been processed yet and all $n$ of its recorded values are the same as those of the currently processed node $N$, the algorithm replaces $N$ by $N'$ in the computation graph. In principle, using larger $n$ will decrease the probability of merging nodes that are not functionally equivalent as long as there is a non-zero chance that any two non-equivalent nodes will have different values (this is the same as the "amplification trick" normally used in the design of randomized algorithms).

It is easy to see that any functionally equivalent nodes will be mapped by the above described algorithm to the same node in the compressed computation graph. However, it can happen that the algorithm will also merge nodes that are not functionally equivalent but just happened (by chance) to output the same values on all the random parameter initializations that the algorithm used. We acknowledge that this can happen in practice, nevertheless it was not commonly encountered in our experiments (Sec. 5), unless explicitly emulated. To do that, we decreased the number of significant digits used in each equivalence check between $value(N; \mathcal{W}_i)$ and $value(N'; \mathcal{W}_i)$. This allows to compress the graphs even further, at the cost of sacrificing fidelity w.r.t. the original model.

There are also cases when we can give (probabilistic) guarantees on the correctness of this algorithm. One such case is when the activation functions in the computation graph are all polynomial. In this case, we can use DeMillo-Lipton-Schwartz-Zippel Lemma (DeMillo & Lipton, 1977) to bound the

---

[2]Here, we add that in this definition, obviously, when we replace a function $f$ by function $f'$, we have to replace all occurrences of $f$ in the graph also by $f'$.

[3]Here, when we say "automorphic nodes", we mean that there exists an automorphism of the graph swapping the two nodes.

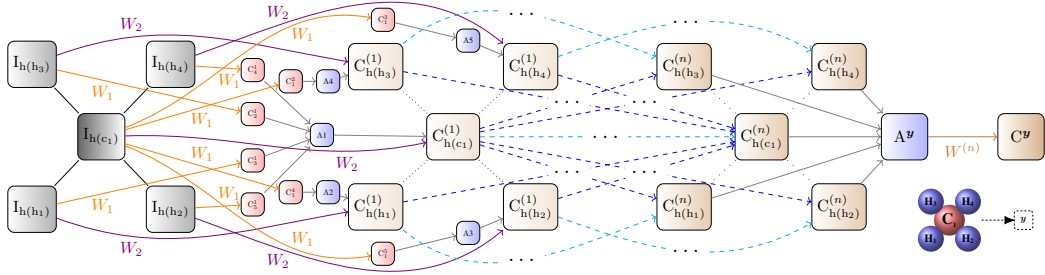

Figure 2: A multi-layer GNN model with a global readout unfolded over an example molecule of methane. Colors are used to distinguish the weight sharing, as well as different node types categorized w.r.t. the associated activation functions, denoted as input (I), convolution (C), and aggregation (A) nodes, respectively.

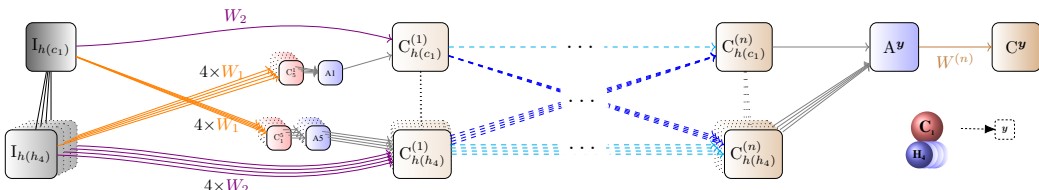

Figure 3: A compressed version of the GNN from Fig. 2, with the compressed parts dotted.

probability of merging two nodes that are not functionally equivalent. However, since the activation functions in the computation graphs that we are interested in are usually not polynomial, we omit the details here. In particular, obtaining similar probabilistic guarantees with activation functions such as ReLU does not seem doable.[4]

## 4.2 AN EXACT COMPRESSION ALGORITHM

The exact algorithm for compressing computation graphs reuses the evaluation with random parameter initializations while recording the respective values for all the nodes. However, the next steps are different. First, instead of traversing the computation graph from the output nodes towards the leaves, it traverses the graph bottom-up, starting from the leaves. Second, rather than merging the nodes with the same recorded value lists right away, the exact algorithm merely considers these as candidates for merging. For that it keeps a data structure (based on a hash table) that indexes the nodes of the computation graph by the lists of the respective values recorded for the random parameter initializations. When, while traversing the graph, it processes a node $N$, it checks if there is any node $N'$ that had the same values over all the random parameter initializations and has already been processed. If so it checks if $N$ and $N'$ are *structurally equivalent* (which we explain in turn) and if they are it replaces $N$ by $N'$. To test the structural equivalence of two nodes, the algorithm checks the following conditions:

1. The activation functions of $N$ and $N'$ are the same.
2. The lists of children of both $N$ and $N'$ are the same (not just structurally equivalent but identical, i.e. *Children(N) = Children(N')*), and if $C$ is the $i$-th child of $N$ and $C'$ is the $i$-th child of $N'$, with $(C, N, L_1)$ and $(C', N, L_2)$ being the respective edges connecting them to $N$, then the labels $L_1$ and $L_2$ must be equal, too.

---

[4]In particular, the proof of DeMillo-Lipton-Schwartz-Zippel Lemma relies on the fact that any single variable polynomial is zero for only a finite number of points, which is not the case for computation graphs with ReLUs.

One can show why the above procedure works by induction. We sketch the main idea here. The base case is trivial. To show the inductive step we can reason as follows. When we are processing the node $N$, by the assumption, the node $N'$ has already been processed. Thus, we know that the children of both $N$ and $N'$ must have already been processed as well. By the induction hypothesis, if any of the children were structurally equivalent, they must have been merged by the algorithm, and so it is enough to check identity of the child nodes. This reasoning then allows one to easily finish a proof of correctness of this algorithm.

There is one additional optimization that we can do for symmetric activation functions. Here by "symmetric" we mean symmetric with respect to permutation of the arguments. An example of such a symmetric activation function is any function of the form $f(x_1, \ldots, x_k) = h\left(\sum_{i=1}^{k} x_k\right)$; such functions are often used in neural networks. In this case we replace the condition 2 above by:

2'. There is a permutation $\pi$ such that $\pi(Children(N)) = Children(N'))$, and if $C$ is the $i$-th child of $N$ and $C'$ is the $\pi(i)$-th child of $N'$, with $(C, N, L_1)$ and $(C', N, L_2)$ being the respective edges connecting them to $N$, then the labels $L_1$ and $L_2$ must be equal.

It is not difficult to implement the above check efficiently (we omit the details for brevity). Note also that the overall asymptotic complexity of compressing a graph $\mathcal{G}$ with either of the algorithms is simply linear in the size of the graph. Specifically, it is the same as the $n$ evaluations of $\mathcal{G}$.

Finally, to illustrate the effect of the lossless compression, we show the GNN model (Sec. 2.1), unfolded over a sample molecule of methane from Fig.2, compressed in Fig. 3.

## 5 EXPERIMENTS

To test the proposed compression in practice, we selected some common structured convolutional models, and evaluated them on a number of real datasets from the domains of (i) molecule classification and (ii) knowledge-base completion. The questions to be answered by the experiments are:

1. How numerically efficient is the non-exact algorithm in achieving lossless compression?
2. What improvements does the compression provide in terms of graph size and speedup?
3. Is learning accuracy truly unaffected by the, presumably lossless, compression in practice?

**Models** We chose mostly GNN-based models as their dynamic computation graphs encompass all the elements of structured convolutional models (convolution, pooling, and recursive layer stacking). Particularly, we choose well-known instances of GCNs and graph-SAGE (Sec. 2.1), each with 2 layers. Additionally, we include Graph Isomorphism Networks (GIN) (Xu et al., 2018a), which follow the same computation scheme with 5 layers, but their particular operations ($C_{W_1} = identity$, $A = sum$, $C_{W_2} = MLP$) are theoretically substantiated in the expressiveness of the Weisfeiler-Lehman test (Weisfeiler, 2006). This is interesting in that it should effectively distinguish non-isomorphic substructures in the data by generating consistently distinct computations, and should thus be somewhat more resistant to our proposed compression. Finally, we include a relational template ("graphlets") introduced in (Sourek et al., 2018), which generalizes GNNs to aggregate small 3-graphlets instead of just neighbors.

**Datasets** For structure property prediction, we used 78 organic molecule classification datasets reported in previous works (Ralaivola et al., 2005; Helma et al., 2001; Lodhi & Muggleton, 2005). Nevertheless, we show only the (alphabetically) first 3 for clarity, as the target metrics were extremely similar over the whole set. We note we also extended GCNs with edge embeddings to account for the various bond types, further *decreasing* the symmetries. For knowledge base completion (KBC), we selected commonly known datasets of Kinships, Nations, and UMLS (Kok & Domingos, 2007) composed of different object-predicate-subject triplets. We utilized GCNs to learn embeddings of all the items and relations, similarly to R-GCNs (Schlichtkrull et al., 2018), and for prediction of each triplet, we fed the three embeddings into an MLP, such as in (Dong et al., 2014), denoted as "KBE".

The size of the individual input graphs is generally smallest in the molecular data with app. 25 atoms and 50 bonds per a single molecule, where there are app. 3000 molecules in each of the datasets

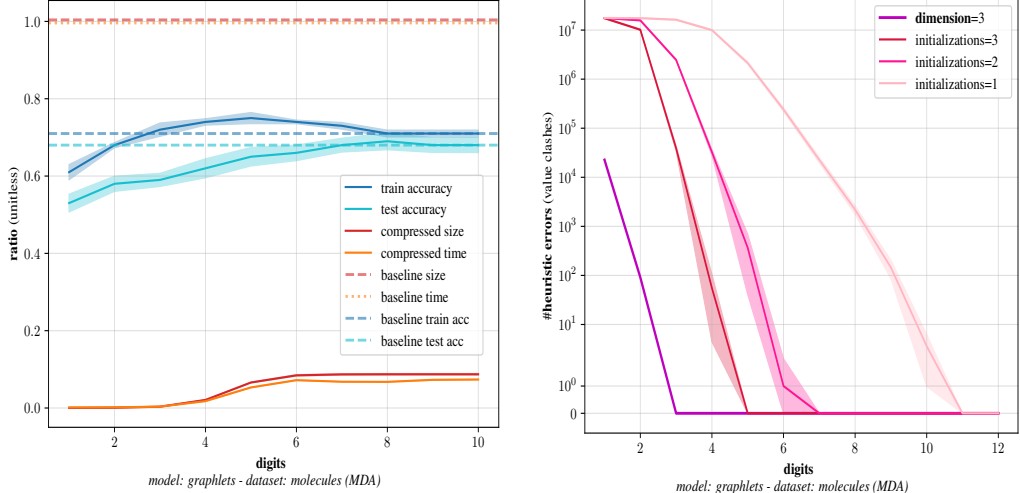

Figure 4: Compression of a *scalar*-parameterized graphlets model on a molecular dataset. We display progression of the selected metrics w.r.t. increasing number of significant digits (inits=1) used in the value comparisons (left), and number of non-equivalent subgraph value clashes detected by the exact algorithm w.r.t. the digits, weight re-initializations, and increased weight dimension (right).

Table 1: Training times *per epocha* across different models and frameworks over 3000 molecules. Additionally, the startup graphs creation time of LRNNs (including the compression) is reported.

| Model | Lifting (s) | LRNNs (s) | PyG (s) | DGL (s) | LRNN startup (s) |
|---|---|---|---|---|---|
| GCN | **0.25 ± 0.01** | 0.75± 0.01 | 3.24 ± 0.02 | 23.25 ± 1.94 | 35.2 ± 1.3 |
| g-SAGE | **0.34 ± 0.01** | 0.89± 0.01 | 3.83 ± 0.04 | 24.23 ± 3.80 | 35.4 ± 1.8 |
| GIN | **1.41 ± 0.10** | 2.84± 0.09 | 11.19 ± 0.06 | 52.04 ± 0.41 | 75.3 ± 3.2 |

on average. The input graphs are then naturally largest for the knowledge bases with app. 20,000 triples over hundreds of objects and relations. The sizes of the corresponding computation graphs themselves are then in the orders of $10^2$–$10^5$ nodes, respectively.

**Experimental Protocol**  We approached all the learning scenarios under simple unified setting with standard hyperparameters, none of which was set to help the compression (sometimes on the contrary). We used the (re-implemented) LRNN framework to encode all the models, and also compared with popular GNN frameworks of PyTorch Geometric (PyG) (Fey & Lenssen, 2019) and Deep Graph Library (DGL) (Wang et al., 2019). If not dictated by the particular model, we set the activation functions simply as $C_W = \frac{1}{1+e^{-W \cdot x}}$ and $A = avg$. We then trained against $MSE$ using 1000 steps of ADAM, and evaluated with a 5-fold crossvalidation.

## 5.1 RESULTS

Firstly, we tested numerical efficiency of the non-exact algorithm itself (Sec. 4), for which we used scalar weight representation in the models to detect symmetries on the level of individual "neurons" (rather than "layers"). We used the (most expressive) graphlets model, where we checked the functional symmetries to overlap with the structural symmetries. The results in Fig. 4 then show that the non-exact algorithm is already able to perfectly distinguish all structural symmetries with but a single weight initialization within less than 12 significant digits. While more initializations indeed improved the efficiency rapidly, in the end they proved unnecessary (but could be used in cases where the available precision would be insufficient). Moreover this test was performed with the actual low-range logistic activations. The displayed (10x) training time improvement (Fig.4 - left) in

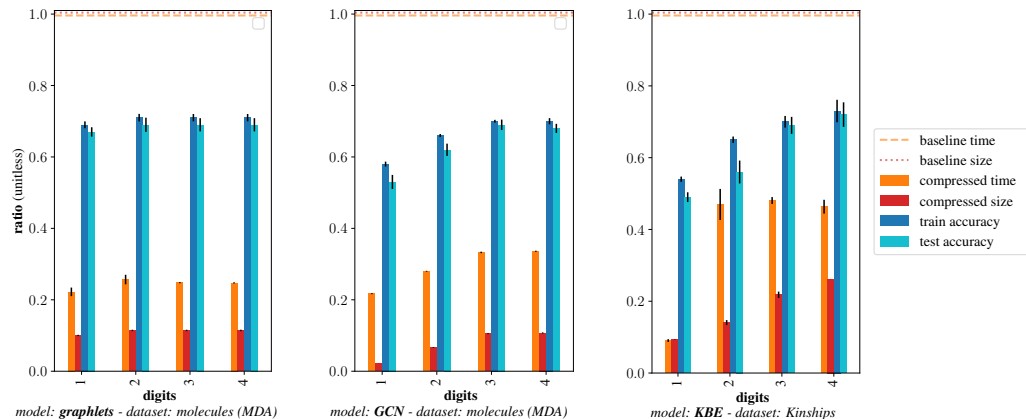

Figure 5: Compression of 3 *tensor*-parameterized models of graphlets (left), GCNs (middle) and KBEs (right) over the molecular (left, middle) and Kinships (right) datasets, with progression of selected metrics against the increasing number of significant digits used for equivalence checking.

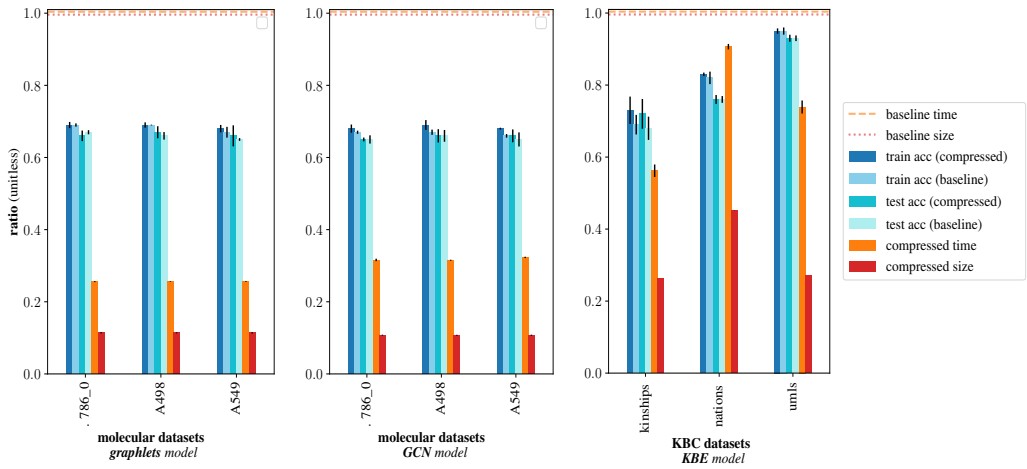

Figure 6: Comparison of 3 different baseline models of graphlets (left), GCNs (middle), and KBEs (right) with their compressed versions over molecule classification (left, middle) and KBC (right).

the scalar models was then directly reflecting the network size reduction, and could be pushed further by decreasing the numeric precision at the expected cost of degrading the learning performance.

Secondly, we performed similar experiments with standard tensor parameterization, where the equivalences were effectively detected on the level of whole neural "layers", since the vector output values (of dim=3) were compared for equality instead. This further improved the precision of the non-exact algorithm (Fig. 4 - right), where merely the first 4 digits were sufficient to achieve lossless compression in all the models and datasets (Figure 5). However, the training (inference) time was no longer directly reflecting the network size reduction, which we account to optimizations used in the vectorized computations. Nevertheless the speedup (app. 3x) was still substantial.

We further compared with established GNN frameworks of PyG (Fey & Lenssen, 2019) and DGL (Wang et al., 2019). We made sure to align the exact computations of GCN, graph-SAGE, and GIN, while all the frameworks performed equally w.r.t. the accuracies. For a more fair comparison, we further increased all (tensor) dimensions to a more common dim=10. The compression effects, as well as performance edge of the implemented LRNN framework itself, are displayed in Tab. 1

for a sample molecular dataset (MDA). Note that the compression was truly least effective for the aforementioned GIN model, nevertheless still provided app. 2x speedup.

Finally, the results in Fig. 6 confirm that the proposed lossless compression via lifting, with either the exact algorithm or the non-exact algorithm with a high-enough numeric precision used, indeed does not degrade the learning performance in terms of training and testing accuracy (both were close within margin of variance over the crossvalidation folds).

Note that the used templated models are quite simple and do not generate any symmetries on their own (which they would, e.g., with recursion), but rather merely reflect the symmetries in the data themselves. Consequently, the speedup was overall lowest for the knowledge graph of Nations, represented via 2 distinct relation types, and higher for the Kinships dataset, representing a more densely interconnected social network. The improvement was then generally biggest for the highly symmetric molecular graphs where, interestingly, the compression often reduced the neural computation graphs to a size even smaller than that of the actual input molecules. Note we only compressed symmetries within individual computation graphs (samples), and the results thus cannot be biased by the potential existence of isomorphic samples (Ivanov et al., 2019), however, potentially much higher compression rates could be also achieved with (dynamic) batching.

## 6    CONCLUSION

We introduced a simple, efficient, lossless compression technique for structured convolutional models inspired by lifted inference. The technique is very light-weight and can be easily adopted by any neural learner, but is most effective for structured convolutional models utilizing weight sharing schemes to target relational data, such as in various graph neural networks. We have demonstrated with existing models and datasets that a significant inference and training time reduction can be achieved without affecting the learning results, and possibly extended beyond for additional speedup.

ACKNOWLEDGMENTS

This work was supported by the Czech Science Foundation project GJ20-19104Y. GS and FZ are also supported by the Czech Science Foundation project 20-29260S.

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
