# OpenReview forum: "Lossless Compression of Structured Convolutional Models via Lifting"
_ICLR.cc/2021/Conference — ICLR 2021 Poster_

### Official Review · AnonReviewer4 · 2020-10-28
**Room for improvement**

**Rating:** 5
**Confidence:** 2

**Review:**

##########################################################################

Summary:


The paper provides an interesting work in the scale/speed up of structured convolutional models. In particular, it proposes an idea using a technique named lifting which is used in scaling up of graphical models to detect the symmetries and compress the neural model such as Graph Neural Network. Authors show that this compression can lead to speedups of the models in many tasks.

##########################################################################

Pros:

1. The paper takes one of the most important issue of structured convolutional models: scale/speed up. I think this is an area with real world implication where we need more works.


2. I think this method can be very useful for networks using lots of identical nodes like molecular network. It can leverage from the repetition in nodes and compress the network into some smaller network and save time and memory.

3. Overall the paper is well written. I liked the illustrations for explaining the methods. Result section is also well structured. It clearly shows the effectiveness of the algorithm over other methods.

##########################################################################

Cons:


1. Although I think this work is good for small datasets like molecular networks, i am skeptic about its transferability in large scale graphs. The reason behind that is I don't assume there would be a lot of structurally similar nodes in case of a large scale real life graph. Authors did not provide any information on the size of the graphs used for the experiments.

2. Authors proposed two different algorithm, exact and non-exact. There are no comparisons between these two methods are shown in the result. Proper analysis on time and performance of the algorithms are also absent. The motivation behind using exact or non-exact is not clear.

3. On Table 1, the time of the different experiments are given. There are no analysis on time vs performance metric in this table. Does all the baseline show similar performance as the baseline model? The analysis is not present.

4. It would also be interesting to see an analysis on the effect of the algorithms on different graph types based on their characteristics (degree distribution, density, veracity etc).

##########################################################################

Questions during rebuttal period:


Please address and clarify the cons above


#########################################################################

---

> ### Author Response · Authors · 2020-11-22
> **Author response**
>
> Thank you for the review!
>
> > *1. Although I think this work is good for small datasets like molecular networks, i am skeptic about its transferability in large scale graphs. The reason behind that is I don't assume there would be a lot of structurally similar nodes in case of a large scale real life graph. Authors did not provide any information on the size of the graphs used for the experiments.*
>
> It is true that the effectiveness of the compression depends on the amount of symmetries in the computation graphs, which stem from the input graph symmetries together with the model symmetries (weight-sharing). It is however not dependent on the size of the graphs.
>
> The individual input graphs vary from the smallest with app. 25 nodes and 50 bonds per a single molecule, to medium-size knowledge bases with app. 20,000 triples over hundreds of objects and relations. We note that the molecular data are not "small datasets", as there are thousands of molecules in each (and so the overall dataset size can be actually bigger than some "large" graphs).
>
> We also note that these are real-life datasets, and the molecular data represent one of the primary GNN applications (Zhou, Jie, et al, 2018).
>
> The computation graphs themselves are then in the orders of 10^2-10^5 of nodes.
>
> We added the information on graph sizes into the paper.
>
> >*2. Authors proposed two different algorithm, exact and non-exact. There are no comparisons between these two methods are shown in the result. Proper analysis on time and performance of the algorithms are also absent. The motivation behind using exact or non-exact is not clear.*
>
> These two algorithms are not to be considered as competing alternatives in terms of performance. The practical difference is that the exact algorithm provides guarantees on lossless compression, while the non-exact does not. However in the experiments, we demonstrate that even the non-exact algorithm achieves lossless compression in practice, given simply a high enough numeric precision to be used in the value comparisons (Fig. 4). The results of the two algorithms are then equivalent in practice.
>
> The non-exact algorithm is slightly faster, since it skips the structural equivalence check, nevertheless the asymptotic complexity of both is the same and is linear in the size of the computation graph (i.e. the same as traversing the graph for simple evaluation - see end of Section 4).
>
> The main motivation behind the exact vs. non-exact algorithms were the structural vs. functional symmetries (see beginning of Section 4). Without having the exact algorithm, we could not show that the non-exact algorithm behaves as an exact one in practice (given sufficiently many bits of precision).
>
> >*3. On Table 1, the time of the different experiments are given. There are no analysis on time vs performance metric in this table. Does all the baseline show similar performance as the baseline model? The analysis is not present.*
>
> Yes, each of the models achieves the same accuracy (performs equivalent computations) across all the frameworks (see Sec. 5.1., p. 8). The computation time is thus all that matters. A more detailed analysis and comparison of the frameworks can be found in the work of (Sourek et. al., 2020).
>
> >*4. It would also be interesting to see an analysis on the effect of the algorithms on different graph types based on their characteristics (degree distribution, density, veracity etc).*
>
> We generally agree, but the compression performance of the algorithm(s) depends on the input graph and model properties (symmetries) in a more complicated manner. Consequently, one could surely select or simulate input graphs of very different properties leading all to extreme compression rates using the right combination of particular node types and patterns of their interconnections extracted by the respective models.
> Instead, in this paper we provide experiments over varying real-world datasets and models from actual practice to reflect the expected real-life performance.
>
>
>
> ------
>
>
> Zhou, Jie, et al. "Graph neural networks: A review of methods and applications." arXiv preprint arXiv:1812.08434 (2018).
>
> Sourek, Gustav, Filip Zelezny, and Ondrej Kuzelka. "Beyond Graph Neural Networks with Lifted Relational Neural Networks." arXiv preprint arXiv:2007.06286 (2020).

---

### Official Review · AnonReviewer2 · 2020-10-28
**An approach to use symmetries to improve scalability of learning in graph based neural networks. The idea seems promising and has been successful in the graphical models community but some details were unclear in the paper.**

**Rating:** 6
**Confidence:** 3

**Review:**

The paper proposes an approach to make learning deep neural networks more efficient using ideas from lifted inference for relational probabilistic models. Specifically, symmetries in the computation graph are identified and then the neural network is compressed into an equivalent model.

The idea of using symmetries for improving scalability has been successfully used in statistical relational models. Therefore, it seems to be a nice direction for improving scalability in neural networks such as GCNs. Thus, large computation graphs can be reduced to smaller graphs for effective computation.

The main weakness with the paper is that the compression algorithm was not very clear to me. Particularly, the problem of identifying isomorphisms is a hard problem and several lifted inference techniques have typically used approximate methods to identify symmetries in graphical models [e.g. Niepert UAI 2012, Bui et. al. UAI 2013, Holtzen et al. UAI 2019, etc.]. I was not sure why or how the identification of exact symmetries is not hard in this case. Is the type of computation graph restricted to some form? If so, a more detailed description of the properties of this graph is useful. Since the main focus of the paper is on “exact” symmetries, I think in general, a more detailed analysis of the proposed approach will help in general.

Regarding the experiments, they show that the proposed techniques can help speed up different types of deep models based on GNNs. One aspect that was not very clear was how large were the computation graphs? Also, when the compression is exact would the accuracies still vary between the compressed and uncompressed model?

In general, I like the idea of using symmetries to compress graph-based neural networks. I am not too clear on the details of the proposed approach, particularly their applicability for general graph structures.

From the discussions, the authors make it clear that "compressing" the computation graph is possible without the need for expensive operations (as is the case in typical "lifted" inference literature). The approach does seem to be simple to implement , maybe a bit more detailed analysis and clarity as suggested by others as well could strengthen the paper further.

---

> ### Author Response · Authors · 2020-11-22
> **Author response**
>
> Thank you for the review!
>
> > *The main weakness with the paper is that the compression algorithm was not very clear to me. Particularly, the problem of identifying isomorphisms is a hard problem and several lifted inference techniques have typically used approximate methods to identify symmetries in graphical models [e.g. Niepert UAI 2012, Bui et. al. UAI 2013, Holtzen et al. UAI 2019, etc.]. I was not sure why or how the identification of exact symmetries is not hard in this case. Is the type of computation graph restricted to some form? If so, a more detailed description of the properties of this graph is useful. Since the main focus of the paper is on “exact” symmetries, I think in general, a more detailed analysis of the proposed approach will help in general.*
>
> The type of the computation graphs is not restricted, but the algorithm does not promise to detect isomorphisms or automorphisms in arbitrary graphs.
>
> The “trick” is that our method does not actually need to be able to detect isomorphism or find automorphisms, which is why it can sidestep the computational complexity obstacles. Automorphisms are important in some other problems (such as probabilistic inference in some of the references you mention) but they are not necessary for compressing computational graphs, as we argue below. The description below applies to the exact algorithm (which is the focus of the reviewer’s comment).
>
> Consider the following computation graph (using the notation used in the paper where we have (node1, node2, edge label)): (1, 3, 1), (2,3,1), (4,5,1), (3,6,1), (5,6,1)
>
> This can be visualized as (v1 -> v3), (v2 -> v3), (v4 -> v5), (v3 -> v6), (v5 -> v6) and all edges are labelled with the same weight w_1.
>
> Suppose that we have the constant activation functions for the leaves of the graph: f_1 = 1, f_2= 2, f_4 = 1. And the other activation functions are, say, a sigmoid.
>
> The output of the exact algorithm will be a graph corresponding to:
>
> (v1 -> v3), (v2 -> v3), (v1 -> v5), (v3 -> v6), (v5 -> v6)
>
> In this case, we only removed the node v4 and replaced it by v1 (so the compression gain is not high but that is not the point of this example). The point of this example is to illustrate that we merged v1 and v4 despite the fact that there is no automorphism mapping v1 to v4 (in the graph theoretical sense).
>
> Hopefully, this example helps to illustrate why we do not have to worry about complexity of detecting symmetries such as graph automorphisms - our algorithm simply does not use them.
>
> We have added a new paragraph explaining in more detail how structural-equivalence differs from detecting automorphism (page 4).
>
>
>
> > *Regarding the experiments, they show that the proposed techniques can help speed up different types of deep models based on GNNs. One aspect that was not very clear was how large were the computation graphs?*
>
> The computation graph sizes are in the orders of 10^2-10^5 of nodes, reflecting the input graphs ranging from small molecules to knowledge bases. We added the information into the paper.
>
>
>
> > *Also, when the compression is exact would the accuracies still vary between the compressed and uncompressed model?*
>
> No, they are practically the same, as one would expect from a lossless compression. This was the very purpose of experiments targeting the proposed experimental Question 3. (p. 6), the results of which are in Figure 6 (and also in Fig. 4 - left).

---

### Official Review · AnonReviewer1 · 2020-10-31

**Rating:** 6
**Confidence:** 1

**Review:**

The authors of this paper propose an compression technique for GNNs that was inspired by lifted inference. The compression consists of removing asymmetries by merging nodes. They define two algorithms for compression: a non-exact algorithm that merges two nodes that are "functional" equivalent and an exact algorithm that merges two nodes that are structurally equivalent.

The approach seem quite novel, pretty interesting and it could interest a wide audience that works on graph models. However, I think the presentation could be improved. In particular, it is not clear the evalutation settings that they are using. For knowledge completion how do they compute accuracy? Do they use golden triples?

In conclusion I think section 5 should be improved and writte more clearly but overall I think it is a good paper and should be considered for acceptance.

---

> ### Author Response · Authors · 2020-11-22
> **Author response**
>
> Thank you for the review!
>
> > *However, I think the presentation could be improved. In particular, it is not clear the evalutation settings that they are using. For knowledge completion how do they compute accuracy? Do they use golden triples?*
>
> We used a standard process of measuring accuracy with golden triples and negative samples generated by corrupting the golden triples (Wang, Quan, et al., 2017). We do not elaborate much on accuracy computation in the paper as the focus here is on computing performance (speedup) and not accuracy of the models.
> I.e. the achieved accuracy values themselves are not important, they are only shown to demonstrate the relative effect of the compression on accuracy (i.e. no effect in the case of lossless compression).
>
>
> Wang, Quan, et al. "Knowledge graph embedding: A survey of approaches and applications." IEEE Transactions on Knowledge and Data Engineering 29.12 (2017): 2724-2743.

---

### Public Comment · ~Thiago_Serra1 · 2020-11-12
**Background on lossless compression**

Regarding the comment in the paper that "to our best knowledge, all the existing techniques are lossy in nature", I would like to point out to the authors some recent work that I did with some colleagues on lossless compression of rectifier networks:

https://arxiv.org/abs/2001.00218

The methods are definitely different, but I believe that it would be adequate to mention our prior contribution to this topic.

---

> ### Author Response · Authors · 2020-11-22
> **related work updated**
>
> Thank you for the reference, we were not aware of this recent work. We agree that is it very different, but shares the same goal of a lossless computation compression. Added to related work.

---

### Decision · Program_Chairs · 2021-01-07
**Final Decision**

**Decision:**

Accept (Poster)

**Comment:**

The paper shows that show that methods for probabilstic lifted inference can also be used to "compress symmetries" in convolutional models over structured data. The resulting structured convolutional models are then shown to yield speed ups for learning graph neural networks, too. This is highly interesting since the existing literature rather considered how to make use of weisfeiler lehman for classification of graphs, both in neural and a kernel way. This paper, however, shows how to compress the computations. And it paves the way to connect equivariant learning lifted inference by exploiting the connection between lifted probabilistic inference / weisfeiler lehman and their algebraic formulations.